# Inter-Species Redox Coupling by Flavin Reductases and FMN-Dependent Two-Component Monooxygenases Undertaking Nucleophilic Baeyer–Villiger Biooxygenations

**DOI:** 10.3390/microorganisms11010071

**Published:** 2022-12-27

**Authors:** Andrew Willetts

**Affiliations:** Curnow Consultancies Ltd., Helston, Cornwall TR13 9PQ, UK; andrewj.willetts@btconnect.com; Tel.: +44-7966-968487

**Keywords:** diketocamphane monooxygenase, luciferase, flavin reductase, two-component monooxygenase, biocatalysis, bioremediation

## Abstract

Using highly purified enzyme preparations throughout, initial kinetic studies demonstrated that the isoenzymic 2,5- and 3,6-diketocamphane mono-oxygenases from *Pseudomonas putida* ATCC 17453 and the LuxAB luciferase from *Vibrio fischeri* ATCC 7744 exhibit commonality in being FMN-dependent two-component monooxygenases that promote redox coupling by the transfer of flavin reductase-generated FMNH_2_ by rapid free diffusion. Subsequent studies confirmed the comprehensive inter-species compatibility of both native and non-native flavin reductases with each of the tested monooxygenases. For all three monooxygenases, non-native flavin reductases from *Escherichia coli* ATCC 11105 and *Aminobacter aminovorans* ATCC 29600 were confirmed to be more efficient donators of FMNH_2_ than the corresponding tested native flavin reductases. Some potential practical implications of these outcomes are considered for optimising FMNH_2_-dependent biooxygenations of recognised practical and commercial value.

## 1. Introduction

Historically, the isoenzymic enantiocomplementary diketocamphane monooxygenases (DKCMOs) from the (*rac*)-camphor-degrading bacterium *Pseudomonas putida* ATCC 17453 [1] and the LuxAB luciferase from the bioluminescent marine bacterium *Vibrio fischeri* ATCC 7744 [2], along with a small number of other known enzymes, were first grouped together on shared functional properties as NADH + FMN-dependent Type 2 Baeyer–Villiger monooxygenases (BVMOs) to distinguish them from the NADPH + FAD-dependent Type 1 BVMOs [3]. While the much more prevalent Type 1 BVMOs have always been recognised as single component true flavoprotein enzymes replete with an active site-bound FAD coenzyme [4], it is now acknowledged that the DKCMOs (2,5-DKCMO [EC 1.14.14.108] and 3,6-DKCMO [EC 1.14.14.155]), and LuxAB luciferase (EC 1.14.14.3) are structurally different, being archetypal members of the recently delineated FMN-dependent two-component monooxygenases (fd-TCMOs, [5]). The defining feature of the fd-TCMOs that excludes them from being classed as true flavoproteins is that they are dependent on deploying fully reduced FMN (FMNH_2_) as a cosubstrate rather than as an active site-bound coenzyme. Significantly, the FMNH_2_ cosubstrate is acquired from one or more distal flavin reductases (FRs [EC 1.5.1.x]). FRs are widely distributed enzymes, a number of which have been purified from various bacteria: the nomenclature and characteristics of some well-studied prokaryotic FRs confirmed to fully reduce FMN effectively are summarised in Table 1. Because reduced flavins can react both directly and indirectly with molecular oxygen to generate highly reactive oxygen species (ROS) that are capable of destroying DNA, lipids, and proteins [6,7,8,9], the mode of functioning of the fd-TCMOs poses a considerable challenge to aerobic organisms that consequently requires a close logistical relationship between the two participating activities [10].

Being similar enzymes, the LuxAB luciferase of *V. fischeri* ATCC 7744 and the isoenzymic DKCMOs of *P. putida* ATCC 17453 share a significant amount of directly equivalent biochemistry. Despite the highly specialised evolved role of dodecanal-deploying LuxAB luciferase in promoting bioluminescence [11], purified preparations of each enzyme can catalyse nucleophilic Baeyer–Villiger oxidations of various aliphatic and alicyclic abiotic ketones to corresponding biooxygenated products [12,13]. However, both isoenzymic DKCMOs do exhibit an important difference from the luciferase with respect to the facial diastereoselectivity of the hydride ion exchanges promoted by the FMNH_2_ cosubstrate (*si*-face versus *re*-face, respectively). This in turn can generate enantiodivergent chiral lactones from racemic and prochiral alicyclic ketone substrates. A number of these products have considerable proven value as synthons in the chemoenzymatic synthesis of sarkomycin A [14], clerodin [15], R-(+)-lipoic acid [16,17], azadirachtin [18], various carbocyclic nucleoside [19] and prostaglandin [20] analogues, plus a significant number of α,ω-dicarboxylic and α-aminocarboxylic acids [21], including the Nylon 6 monomer 6-aminohexanoic acid [22] and the Nylon 9 monomer 9-aminononanoic acid [23]. Another shared characteristic of the luciferase and the DKCMOs is that the requisite FMNH_2_ can be generated by multiple alternative native FRs (FRG_Vf_ and Fre_Vf_ in *V. fischeri*, [24]; Frp1, Frp2, Fred, and putidaredoxin reductase (PdR) in *P. putida* ATCC 17453 [10]), which may represent an evolved solution to the problem posed by the relatively inefficient reduction of flavins by reduced pyridine nucleotides [25,26,27,28]. Whereas Frp1, Frp2, Fred, and Fre_Vf_ are devoid of any bound flavin and sequentially deploy FMN and NADH as cosubstrates to generate FMNH_2_, putidaredoxin and FRG_Vf_ are flavoproteins that contain a bound flavin coenzyme and generate FMNH_2_ by a ping-pong reaction mechanism (Figure 1).

These two functionally different types of enzyme have been termed Class I and Class II FRs, respectively [26,28]. That the biochemistry common to the LuxAB luciferase of *V. fischeri* and the isoenzymic DKCMOs of *P. putida* ATCC 17453 may extend to the functional interchangeability of the relevant FMNH_2_-generating and FMNH_2_-oxidising activities of these particular fd-TCMOs was suggested by some preliminary data indicating that one or more of the FRs present in a commercial grade crude extract prepared from cells of *V. fischeri* (Boehringer Mannheim Co) can support the biooxygenation of alicyclic ketones to lactones by purified preparations of each DKCMO isoenzyme [13,29]. The possibility that this apparent compatibility results from the effective functional inter-changeability of discrete highly purified preparations of the relevant FMNH_2_-generating and FMNH_2_-oxidising activities of the DKCMOs and LuxAB luciferase remains to be proven. This was the specific initial goal of the present paper.

Significant in the context of the inter-species compatibility of different functional activities of fd-TCMOs is that the LuxAB luciferase of *V. fischeri* [30,31] and both DKCMOs of *P. putida* [32,33,34] have all proved to be highly active when expressed in recombinant bacteria that are deficient in any of the corresponding native FRs (FRG_Vf_, and Fre_Vf_ [*V. fischeri*]; Frp1, Frp2, Fred, and PdR [*P. putida*]). Consequently, the study has been widened to investigate the ability of purified preparations of the characterised FRs of some other known FMN-dependent two-component enzymes (Table 1) to support the nucleophilic biooxygenating activities of the DKCMOs from *P. putida* ATCC 17453 and the LuxAB luciferase from *V. fischeri* ATCC 7744. This extended aspect of inter-species biocompatibility became the second specific goal of the present paper.

By way of direct support for this wider study, it has been demonstrated that a purified preparation of Fre_Ec_, the major FR isolated from *E. coli*, supports bioluminescence in vitro with the LuxAB luciferase from *V. harveyi* [35]. Less well characterised is recognition that a commercial grade crude extract of FRs prepared from cells of *V. fischeri* (Boehringer Mannheim Co) can support corresponding biooxygenations by the FMNH_2_-dependent monooxygenase moieties of the characterised fd-TCMOs nitrilotriacetate monooxygenase [36], pristinamycin PIIA synthase [37], and EDTA monooxygenase [38]. A relevant related observation is that purified 2,5-DKCMO from ATCC 17453 has been shown to function effectively with FMNH_2_ generated by hydride donation from the synthetic nicotinamide coenzyme biomimetics 1-benzyl-1,4-dihydronicotinamide (BNAH), and 1-(2-carbamoylmethyl)-1,4-dihydronicotinamide (AmNAH) [39].

## 2. Materials and Methods

### 2.1. Bacterial Strains, Culture Maintenance, and Growth Conditions

*P.putida* ATCC 17453 was cultured at 30 ^o^C using a mineral salts medium supplemented with 17.5 mM (*rac*)-camphor as the principal carbon source as fully described previously [27].

*A. aminovorans* ATCC 29600 was cultured at 30 °C using a mineral salts medium supplemented with 1 g L^−1^ of nitrilotriacetate as fully described previously [40].

*Escherichia coli* ATCC 11105 was cultured at 37 °C using LB medium (Sigma-Aldrich, Dorset, UK) as fully described previously [41].

*Vibrio fischeri* ATCC 7744 was cultured at 26 °C using Photobacterium medium (Difco Laboratories, Detroit, Mich., USA) as fully described previously [2].

### 2.2. Extract Preparation

Cultures were grown into mid log phase of growth and the biomass content harvested by centrifugation (10,000× *g* for 15 min at 5 °C [MSE Coolspin 2, MSE, Heathfield, East Sussex, UK]), washed with an equal volume of cold Tris-HCl buffer (0.1 M, pH 7.0) and then recentrifuged. The recovered cells were evenly suspended in 7.5 mL of the same buffer and subsequently sonicated (Soniprep150, MSE, Heathfield, UK) in ice for 3 × 2 min. The resultant homogenates were centrifuged (20,000× *g* for 15 min at 5 °C) to remove the cell debris.

### 2.3. Purification of 2,5-DKCMO, 3,6-DKCMO, and LuxAB Luciferase 

Samples of the highly purified biooxygenating subunits of both enantiocomplementary DKCMO isoenzymes were prepared at 4 °C using a BioLogic FPLC system (BioLogic 10, Bio-Rad, Hercules, CA, USA) deploying anion-exchange chromatography (Q-Sepharose, Sigma-Aldrich, Dorset, UK), followed by a Mono-Q column (Pharmacia, Stockholm, Sweden) eluted with a linear gradient of 0–0.6 M KCl in 21 mM phosphate buffer pH 7.1, as fully described previously [18]. The A_280_ trace from the Mono-Q column output is shown in Appendix A. The isolated 2,5-DKCMO activity comprises an approximately equimolar mixture [12] of 2,5-DKCMO-1 (coded for by the *camE_25-1_* gene) and 2,5-DKCMO-2 (coded for by the *camE_25-2_* gene). Purified DKCMO activities were assayed in the co-presence of 20mU of a commercial FR preparation (NAD(P)H: FMN oxidoreductase from *V. fischeri*, (Boehringer Mannheim, Indianapolis, Ind, USA) by measuring the rate of NADH-stimulated lactone formation from the corresponding diketocamphane substrate by GC (Shimadzu GC-14A, Shimadzu Europe) using a 10% Carbowax 20 M column [27]. Samples of LuxAB luciferase were prepared using the BioLogic FPLC system deploying anion-exchange chromatography (DEAE-Sephadex, Sigma-Aldrich, Dorset, UK) as fully described previously [42]. LuxAB luciferase activity was assayed in the co-presence of 20 mU of a commercial FR preparation (NAD(P)H: FMN oxidoreductase from *V. fischeri* (Boehringer Mannheim, Indianapolis, Ind) by measuring the rate of NADH-stimulated dodecanoic acid formation from 2-tridecanone by GC (Shimadzu GC-14A) using a BP1 column [13].

### 2.4. Purification of Frp1, Frp2, Fre_Ec_, FRG_Vf_, Fre_Vf_, and FRD_Aa_

Samples of highly purified Frp1 and Frp2 were prepared at 4 ^o^C using a BioLogic FPLC system (BioLogic 10, Bio-Rad, Hercules, CA, USA) deploying successive anion-exchange (Mono-Q, Pharmacia, Stockholm, Sweden), affinity (Reactive Blue-4-agarose, Pharmacia), and gel filtration (HiLoad 16/60 Superose 12, Pharmacia) columns in the three-stage protocol fully described previously [27]. Samples of highly purified Fre_Ec_ were prepared using the BioLogic FPLC system deploying a nickel affinity (Sigma-Aldrich) column as fully described previously [35]. Samples of highly purified Fre_Vf_ and FRG_Vf_ were prepared using the BioLogic FPLC system deploying successive gel filtration (Sephadex G-100, Sigma-Aldrich), and anion-exchange (DEAE-Sephadex, Sigma Aldrich) columns as fully described previously [43]. Samples of highly purified FRG_Aa_ were prepared using the BioLogic FPLC system deploying successive anion-exchange (Q-Sepharose [Sigma-Aldrich]), followed by Mono-Q [Pharmacia, Stockholm, Sweden]) and gel filtration (Superdex 75 [Sigma-Aldrich]) columns as fully described previously [40]. SDS-PAGE showing purification of each of the FRs is shown in Appendix A: lane 1, Fre_Vf_; lane 2, FRG_Vf_; lane 3, Frp1; lane 4, Frp2; lane 5, Fre_Ec_; lane 6, FRD_Aa_; lane 7, 2,5-DKCMO; lane 8, 3,6-DKCMO; lane 9, Pharmacia low MW markers.

### 2.5. Single-Enzyme Kinetic Studies

Assays using 20 mU aliquots of highly purified FR enzyme preparations were conducted spectrophotometrically at 340 nm under anaerobic conditions by measuring the initial rate of enzyme-catalysed reduction of FMN by NADH as fully described previously [10].

### 2.6. Coupled-Enzyme Kinetic Studies

Assays using appropriate combinations of 20 mU aliquots of highly purified FR preparations and 200 mU aliquots of highly purified monooxygenase preparations plus 1 mM (*rac*)-bicyclo[3.2.0]hept-2-en-6-one as the biooxidisable substrate were conducted spectrophotometrically at 340 nm by measuring the initial rate of enzyme-catalysed reduction of FMN by NADH as fully described previously [10]. Each reaction mixture was supplemented with 60 U catalase (Sigma-Aldrich) to avoid accumulation of hydrogen peroxide resulting from substrate-independent oxygen consumption.

### 2.7. Longer-Term (120 min) Biocatalytic Reactions with Combinations of Highly Purified Enzymes

Biotransformations with the various combinations of highly purified FRs and monooxygenases were carried out in reaction mixtures (1 mL) containing Tris/HCl buffer (60 mM, pH 7.6), 0.1 mM NADH, 0.03 mM FMN, 60 U catalase, 30 mU formate dehydrogenase, 50 mM sodium formate, 20 mU FR, 200 mU monooxygenase, and 1 mM (*rac*)-bicyclo[3.2.0]hept-2-en-6-one. All biotransformation reaction mixtures were incubated at 25 °C for 120 min, and then assayed by chiral capillary GC using a Lipadex D fused silica column as fully described previously [44,45].

### 2.8. Reproducibility

Where indicated, procedures were repeated three times with equivalent purified enzyme preparations, and the resultant data presented graphically with corresponding standard deviation error bars (Appendix A).

## 3. Results and Discussion

As prior studies with both the isoenzymic DKCMOs from *P. putida* ATCC 17453 [32,33,34,46] and the LuxAB luciferase from *V. fischeri* ATCC 7744 [13] have confirmed that highly purified preparations of each enzyme can biooxygenate (*rac*)-bicyclo[3.2.0]hept-2-en-6-one to some extent, thereby generating a mixture of regioisomeric 2-oxa- and 3-oxa-lactones (2,5-DKCMO > 3,6-DKCMO > LuxAB luciferase), this bicyclic ketone was used as the consensus substrate of choice in the present study of these three fd-TCMO enzymes. An important value of this substrate is that the nature of the ensuing lactone products serves as a monitor of both the regio- and stereoselectivity of the three enzymes. In this respect, prior research with this abiotic bicyclic ketone has indicated that whereas the two enzymic DKCMOs exhibit stereochemical congruence, as clearly reflected by the equivalence of the predominant lactones formed [12], the LuxAB luciferase exhibits stereochemical divergence from both DKCMOs by generating corresponding lactones in the opposite enantiomeric series [13]. However, because both the luciferase and the 3,6-DKCMO isoenzyme exhibit some form of toxic response to the ketone and/or generated ROS which begins to be progressively significant 3–4 hours after initial exposure [13,17], all the current studies that deployed this substrate were conducted over the shorter time-frame of 120 minutes, when typically 70–90%, 40–60%, and 20–40% of the (*rac*)-ketone had been converted to a mixture of the corresponding 2-oxa and 3-oxa lactones by 2,5-DKCMO, 3,6-DKCMO and the LuxAB luciferase, respectively.

Prior to investigating any aspect of the interchangeability of FMNH_2_-generating activities between different fd-TCMOs, a series of relevant comparative kinetic studies were conducted to establish whether the transfer of FR-generated FMNH_2_ to the corresponding monooxygenase moiety of the isoenzymic DKCMOs of *P. putida* and the LuxAB luciferase of *V. fischeri* occurs by rapid free diffusion or involves the formation of some form of transitory complex [47]. The specific aim of these kinetic studies was to establish the apparent *K_m_*_FMN_ values for representative FRs of *P. putida* ATCC 17453 (Fpr1, and Frp2), and *V. fischeri* ATCC 7744 (FRG_Vf_ and Fre_Vf_), when assayed either as a highly purified FR preparation catalysing a single-enzyme FMNH_2_-generating step, or as a coupled-enzyme combination promoting FMNH_2_-turnover in the co-presence of (*rac*)-bicyclic ketone and a 10-fold excess of highly purified preparations of the corresponding native monooxygenase. While the single-enzyme assays were conducted under anaerobic conditions, the corresponding monooxygenase-dependent coupled-enzyme reactions were assayed aerobically, but with the inclusion of catalase to decompose any hydrogen peroxide generated abiotically from FMNH_2_ by molecular oxygen [10]. In each case, FMN was tested over a range of concentrations up to 30 μM, which resulted in the corresponding *v* values asymptotically approaching a consistent *V_max_* value. The outcomes of the representative comparative single- vs. coupled-enzyme kinetic studies, when plotted as corresponding Michaelis–Menten plots (Appendix A), confirmed that in each case the apparent *V_max_* of each coupled-enzyme assay was approximately 2–3-fold lower than that of the corresponding single-enzyme assay, most likely reflecting known differences in the apparent turnover rates of the relevant enzymes [27,41,47]. However, most significantly, for each comparative single- vs. coupled-enzyme study, the corresponding calculated apparent *K_m_*_FMN_ values derived in each case using the Michaelis–Menten equation and relevant mean values were remarkably similar, as summarised in Table 2.

This commonality of the derived kinetic outcomes indicates that 2,5-DKCMO, 3,6-DKCMO, and the LuxAB luciferase of *V. fischeri* each operate a common mode of action that involves the transfer of FR-generated FMNH_2_ by rapid free diffusion, as the alternative formation of a FR-monooxygenase transitory complex involving protein–protein interaction would have been signalled by significantly lower recorded *K_m_*_FMN_ values for the coupled-enzyme reactions (≤1 order of magnitude [41]). Equivalent less rigorously supported proposals were made for the LuxAB luciferase of *V. harveyi* when functioning with the with Fre_Ec_, the major FR of *E. coli* [35], and for the transfer to 2,5-DKCMO of FMNH_2_ generated by the synthetic nicotinamide coenzyme biomimetic BNAH [39].

That rapid free diffusion of the FMNH_2_ cosubstrate from competent native donor FRs is the proven mode of action of all three tested fd-TCMOs may enable them to function effectively with non-native FMNH_2_-generating FRs. This possibility was examined by monitoring the outcomes of 120 min biotransformations of (*rac*)-bicyclo[3.2.0]hept-2-ene -6-one by highly purified preparations of both isoenzymic DKCMOs and the LuxAB luciferase in relevant coupled-enzyme assays with in each case a range of both corresponding native and non-native FRs (Figure 2). The respective relevant highly purified native FRs deployed were those tested previously (Table 2), whereas for all three monooxygenases the two additional non-native NADH-dependent FRs that were deployed were FRD_Aa_ and Fre_Ec_, which both deploy a sequential mechanism of hydride transfer (Table 1). Both FRD_Aa_ and Fre_Ec_ are highly effective NADH-dependent FMN-dependent reductases as reflected by their reported *Km*_FMN_ single-enzyme assay values of 1.0 µM [40] and 0.8 µM [35], respectively. Each biotransformation was supplemented with a formate/formate dehydrogenase NADH-regenerating system to enhance hydride ion availability, and catalase to avoid accumulation of hydrogen peroxide resulting from substrate-independent oxygen consumption. All the tested FRs are NADH-dependent enzymes which, with the exception of FRG_Vf_, deploy a sequential reaction mechanism (Table 2).

Analysis of the stopped 120-minute reaction mixtures (Table 3) indicated a number of shared and monooxygenase-specific outcomes. Most significantly, with respect to redox coupling, the data confirm that the LuxAB luciferase and both isoenzymic DKCMOs can biooxygenate the bicyclic ketone by sourcing the requisite FMNH_2_ cosubstrate from both non-native as well as corresponding native FRs. Because equivalent aliquots of FR (20 mU) and highly purified monooxygenase (200 mU) were used in each case, the residual ketone remaining in the 120-minute stopped reaction mixtures can be taken as a reflexion of the resultant activity of the relevant coupled-enzyme reactions. Relevant in this respect is that the affinity of each monooygenase for FMNH_2_ will be an idiosyncratic constant [5]. The data indicate that for each of the tested fd-TCMOs, the relative order of effectiveness of the alternative highly purified FRs serving as suppliers of FMNH_2_ was Fre_Ec_ > FRD_Aa_ > Frp2 > Frp1 = Fre_Vf_ > FRG_Vf_. This outcome, indicating that Fre_Ec_ and FRD_Aa_ promote the highest recorded levels of ketone biotransformation with each set of coupled-enzyme reactions, correlates with the corresponding *K_m_*_FMN_ values of the tested FRs calculated from relevant single-enzyme kinetic assays listed in Table 2, albeit the values for Fre_Ec_ and FRD_Aa_ (0.8 μM, and 1.0 μM, respectively) are published precedents generated under slightly different reaction conditions. The relatively poor performance of FRG_Vf,_ the only tested FR that deploys a ping-pong rather than a sequential reaction mechanism to generate FMNH_2_ (Figure 1), may be related to the comparatively low *V_max_* recorded for this enzyme (Appendix A). It may also be relevant that for each coupled-enzyme partnership there will be an uncharacterised effect resulting from any proximity factor that may influence the logistics of the transfer of FMNH_2_ by rapid free diffusion between the two participating biocatalysts [5].

Previous studies of enzyme catalysed Baeyer–Villiger reactions have proposed that monooxygenase-dictated stereoelectronic effects serve as the principal determinant in controlling multiple aspects of the selectivity expressed by the relevant outcomes [48,49]. Detailed analysis of the lactones formed from (*rac*)-bicyclo-[3.2.0]hept-2-en-6-one in the currently analysed coupled-enzyme reactions (Table 3) shows that the relative amounts (and consequently the ratio) of the ketone enantiomers biooxidised, and both the absolute configuration and enantiomeric excess of the resultant predominant regioisomeric lactones are highly consistent with all six partner FRs tested in each individual monooxygenase-catalysed series. These data both concur with the proposed cardinal role of the monooxygenases in delineating all currently investigated aspects of their specificity, and conversely confirm a predictable absence of any relevant specificity-related influence(s) resulting from the deployment of the tested range of non-native as well as native coupled-enzyme partner FRs.

The particular outcomes of the biooxygenations catalysed by LuxAB luciferase are interesting in another respect. Because the (−)-(*1S*,*5R*)-2-oxa-lactone formed is an acknowledged synthon for the chemoenzymatic synthesis of various potentially useful prostaglandin analogues [50], it is significant that the calculated enantiomeric purities [45] of the (−)-2-oxa-lactone recorded with this monooxygenase in combination with each of the tested purified FRs are all higher than those reported previously for equivalent biotransformations undertaken by the Type 1 BVMOs 2-oxo-Δ^3^-4,5,5-trimethylcyclopentenylacetyl-CoA monooxygenase sourced from *P. putida* ATCC 17453 [4,45,51], and cyclohexanone monooxygenase sourced from either *Acinetobacter* TD63 [52], or *Acinetobacter calcoaceticus* NCIMB 9871 [53,54].

Clearly, under the reaction conditions deployed, using highly purified enzyme preparations supplemented with catalase to promote hydrogen peroxide scrubbing, NADH-dependent Fre_Ec_ from *E. coli* and FRD_Aa_ from *A. aminovorans* are superior suppliers of FMNH_2_ than each of the corresponding native FRs for each of the tested monooxygenases. Interestingly, purified 2,5-DKCMO plus catalase has also been shown to function effectively in undertaking biooxygenation of (*rac*)-bicyclo[3.2.0]hept-2-en-6-one with FMNH_2_ generated by hydride donation from synthetic nicotinamide coenzyme biomimetics [39]. Although less extensively characterised, an equivalent outcome conducted without supplementary exogenous catalase addition was reported for flavin reductase-coupling with DszA and DszC, two fd-TCMOs involved in the 4S pathway of dibenzothiophene desulfurisation by *Rhodococcus erythropilis* D-1 [55]. Coupled-reactions by both monooxygenases with a non-native FR sourced from *Paenibacillus polymyxa* A-1 were reported to proceeded more efficiently than with the corresponding native FR DszD. Less well defined are a number of reports confirming relevant biooxidative activity in recombinant strains of Fre_Ec_-containing *E. coli* expressing 2,5-DKCMO [32], and 3,6-DKCMO [33,34], and co-expressing both 4S pathway monooxygenases (DszA and DszC) plus the FRGVh gene sourced from *V. harveyi* [56,57,58].

The major outcome of the study has been to use highly purified enzyme preparations to demonstrate the comprehensive inter-species compatability of the FR and monooxygenase componenents of three fd-TCMOs catalysing FMNH_2_-dependent Baeyer–Villiger-type nucleophilic biooxygenations of a bicyclic ketone to corresponding 2-oxa- and 3-oxa-lactones. This, along with another equivalent precedent involving an abiotic source of the requisit FMNH_2_ [39], suggests that FMN-dependent TCMOs are characterised by an evolved commonality of function in deploying the reduced form of the flavin cosubstrate irrespective of its biotic or abiotic source. For all three tested fd-TCMOs, it is most significant that the non-native FRs Fre^Ec^ and FRD^Aa^ were confirmed to be more effective donators of FMNH_2_ than the corresponding native FRs when tested with highly purified enzyme preparations in the co-presence of exogenous catalase, as reflected consistently by all tested parameters of the corresponding coupled reactions. This confirms absolutely a promiscuous aspect to the functioning of these enzymes, as proposed previously for the functioning of both isoenzymic DKCMOs with multiple confirmed competent native FRs (Frp1, Frp2, Fred, and PdR [1]), and specifically for 2,5-DKCMO with the abiotic synthetic nicotinamide biomimetics BNAH and AmNAH [39]. Catalytic promiscuity as a concept has been proposed to serve an important role in enzyme evolution [59,60,61], which in the case of the tested fd-TCMOs may be related to the known inefficiency of FMN reduction by NAD(P)H_2_ [25,26,27]. While the implications of this particular outcome may have considerable potenial practical value for optimising fd-TCMO-catalysed biotransformation and bioremediation processes [3,12,16,17,62,63,64,65,66], this must be tempered with the recognition that plural detrimental toxic effects can result from the production of hydrogen peroxide and superoxide radicals generated by high levels of FR activity in the absence of corresponding levels of catalase [6,7,8,9,67].

## Figures and Tables

**Figure 1 microorganisms-11-00071-f001:**
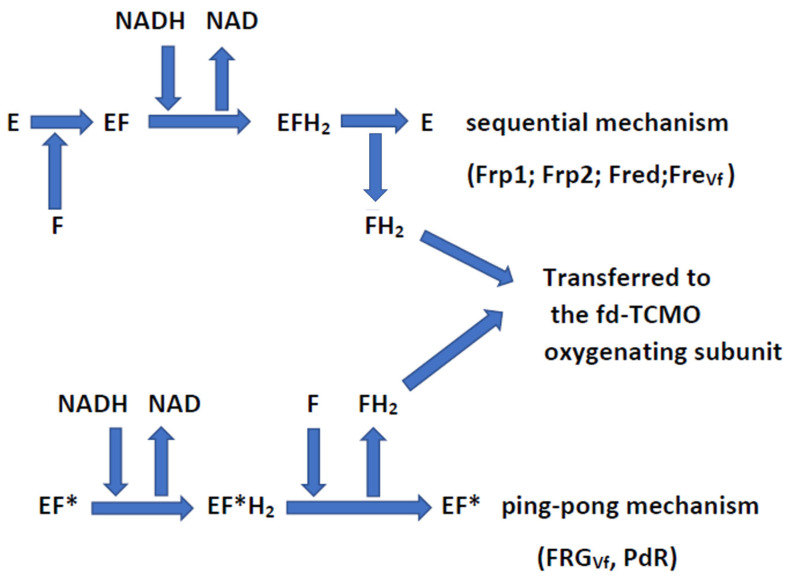
The different types (bound vs. unbound flavin) and the different reaction mechanisms (sequential vs. ping-pong) of the flavin reductases of *V. fischeri* ATCC 7744 and *P. putida* ATCC 17453. E = flavin reductase with unbound flavin coenzyme (Frp1; Frp2; Fred; Fre_Vf_): F = FMN: EF* = flavin reductase with bound flavin coenzyme (FRG_Vf_ [FMN]; PdR [FAD]): FH_2_ = FMNH_2_.

**Figure 2 microorganisms-11-00071-f002:**
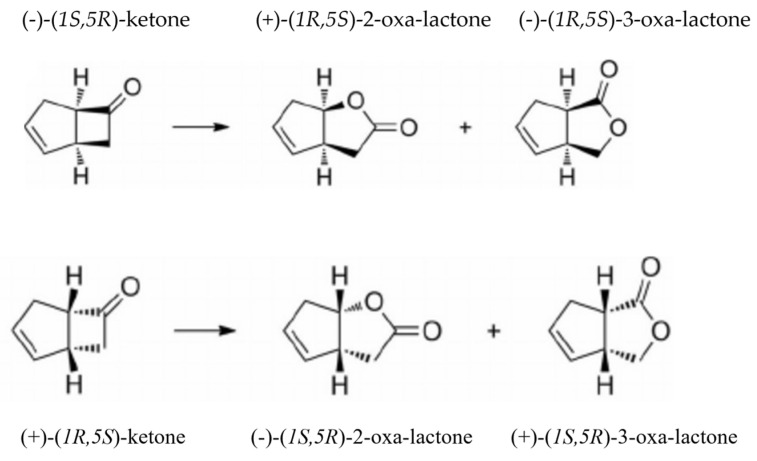
Potential outcomes for the biooxygenation of (*rac*)-bicyclo-[3.2.0]hept-2-en-6-one to 2-oxa- and 3-oxa-lactones by fd-TCMOs dependent on FMNH_2_ sourced from an FR. Tested fd-TCMOs: 2,5-DKCMO; 3,6-DKCMO; LuxAB luciferase. Tested FRs: Frp1 (*P. putida*); Frp2 (*P. putida*); Fre_Vf_ (*V. fischeri*); FRG_Vf_ (*V. fischeri*); Fre_Ec_ (*E. coli*); FRD_Aa_ (*A. aminovorans*).

**Table 1 microorganisms-11-00071-t001:** Characterised FMN-dependent FRs [5].

Flavin Reductase: Nomenclature and Reaction Mechanism	Microbial Source	MW and Structure	Bound or Unbound FMN
Frp1 Sequential	*P. putida*	26.0 kDa	Unbound
Frp2 Sequential	*P. putida*	27.0 kDa	Unbound
Fred Sequential	*P. putida*	2 × 18.0 kDa	Unbound
PdR Ping-pong	*P. putida*	48.5 kDa	Bound
Fre_Ec_ (FR-II) Sequential	*E. coli*	28.5 kDa	Unbound
FRG_Vf_ (FRG-I) Ping-pong	*V. fischeri*	24.6 kDa	Bound
Fre_Vf_ Sequential	*V. fischeri*	25.5 kDa	Unbound
FRD_Vh_ Sequential	*V. harveyi*	26.7 kDa	Unbound
FRP_Vh_ Ping-pong	*V. harveyi*	26.3 kDa	Bound
ActVB (FRD-II) Ping-pong	*Streptomyces coelicolor*	2 × 18.0 kDa	Bound
SnaC (FRD-I, PII_B_) Ping-pong	*S. pristinaspiralis*	28.0 kDa	Bound
EmoB (cB’) Ping-pong	*Chelatovorans* *multitrophicus*	2 × 25.0 kDa	Bound
DszD (FRD-III) Sequential	*Rhodococcus* *erythropolis*	4 × 22.5 kDa	Unbound
FRD_Aa_ (cB) Sequential	*Aminobacter* *aminovorans*	2 × 44.0 kDa	Bound

**Table 2 microorganisms-11-00071-t002:** Calculated apparent *Km*_FMN_ values for highly purified representative FRs from *P. putida* ATCC 17453 and *V. fischeri* ATCC 7744 tested both as single-enzyme assays (S) and coupled-enzyme assays with **2,5-DKCMO** (C:+**2,5-MO**), **3,6-DKCMO** (C**:+3,6-MO**), and **LuxAB luciferase** (C:+**LuxAB**) and, in each case, 1mM of the biooxidisable ketone (*rac*)-bicyclo[3.2.0]hept-2-en-6-one. The equivalent reported single-enzyme *Km*_FMN_ values for Fre_Ec_ and FRD_Aa_ are 0.8 μM [35] and 1.0 μM [40], respectively.

Flavin Reductase and Source	Apparent *Km*_FMN_ (μM)	Source of Hydride Ion	Mechanism of Hydride Ion Transfer
Frp1 *P. putida*	2.5 (S) 2.0 (C + **2,5-MO**) 2.5 (C + **3,6-MO**)	NADH	Sequential
Frp2 *P. putida*	4.2 (S) 3.6 (C + **2,5-MO**) 4.1 (C + **3,6-MO**)	NADH	Sequential
FRG_Vf_ (*V. fischeri*)	4.3 (S) 4.0 (C + **LuxAB**)	NADH	Ping-pong
Fre_Vf_ (*V. fischeri*)	2.5 (S) 2.6 (C + **LuxAB**)	NADH	Sequential

**Table 3 microorganisms-11-00071-t003:** 120 min outcomes of the fd-TCMO-FR coupled-enzyme reactions with highly purified preparations of **LuxAB luciferase**, **2,5-DKCMO**, and **3,6-DKCMO** with FRs from *P. putida* (Frp1 and Frp2), *V. fischeri* (FRG_Vf_ and Fre_Vf_), *A. aminovorans* (FRD_Aa_) and *E. coli* (Fre_Ec_). The structures of (+)k, (−)k, (+)2l, (−)2l, (+)3l, and (−)3l are shown in Figure 2. (n) = native FR, (n.n) = non-native FR.

fd-TCMO and Partner FR in Coupled-enzyme Reaction		Residual Ketone (mM) Remaining after 120 min	Percentage and Ratio of the Ketone (k) Enantiomers Biotransformed			Predominant Regioisomeric Lactones (l) Formed Expressed as ee%			
			(+)k	(−)k	(+):(−)	(+)2l	(−)2l	(+)3l	(−)3l
LuxAB luciferase									
	Frp1 (n.n)	0.70	41.6	18.4	2.26:1	–	>98	–	16
	Frp2 (n.n)	0.69	43.0	18.4	2.33:1	–	>98	–	18
	FRG_Vf_ (n)	0.75	34.2	15.0	2.28:1	–	96	–	18
	Fre_Vf_ (n)	0.79	42.8	18.8	2.28:1	–	>99	–	16
	FRD_A_ (n.n)	0.62	52.2	22.8	2.29:1	–	>98	–	16
	Fre_Ec_ (n.n)	0.61	54.2	24.0	2.25:1	–	>99	–	18
2,5-DKCMO									
	Frp1 (n)	0.24	81.8	69.7	1.17:1	84	–	>99	–
	Frp2 (n)	0.20	84.3	74.8	1.13:1	86	–	>98	–
	FRG_Vf_ (n.n)	0.26	78.8	68.4	1.15:1	82	–	>98	–
	Fre_Vf_ (n.n)	0.23	82.5	71.7	1.15:1	84	–	>98	–
	FRD_Aa_ (n.n)	0.06	100	88.6	1.13:1	88	–	>99	–
	Fre_Ec_ (n.n)	0.04	100	92.0	1.09:1	90	–	>98	–
3,6-DKCMO									
	Frp1 (n)	0.53	58.8	35.0	1.68:1	30	–	80	–
	Frp2 (n)	0.53	59.8	35.2	1.70:1	30	–	82	–
	FRG_Vf_ (n.n)	0.55	57.0	33.0	1.72:1	32	–	86	–
	Fre_Vf_ (n.n)	0.52	61.0	35.4	1.72:1	32	–	84	–
	FRD_Aa_ (n.n)	0.38	77.6	45.0	1.72:1	34	–	82	–
	Fre_Ec_ (n.n)	0.37	80.0	45.8	1.74:1	32	–	88	–

## Data Availability

The original contributions presented in the study are included in the article: further enquiries can be directed to the corresponding author.

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
