# Peer review of "Inter-Species Redox Coupling by Flavin Reductases and FMN-Dependent Two-Component Monooxygenases Undertaking Nucleophilic Baeyer–Villiger Biooxygenations"

_microorganisms, 2022, doi:10.3390/microorganisms11010071_

Round 1

Reviewer 1 Report

In this work, the author compared the kinetic properties of three fd-TCMOs in the presence of different flavin reductases and found that FMNH2 transfers via free diffusion. In addition, non-native FRs serve as more efficeint FMNH2 donors than native ones. While the manuscript proposes an interesting hypothesis, I think more experiments are needed to support it.

1. In the introduction, the author presented thorough background information. However, it would be helpful to also include the purpose of this study in order for readers to understand more easily.

2. Line 183:  (2,5-DKCMO>3,6-DKCMO>LuxAB luciferase), what is being compared here? Is it the amount of lactone produced, enzyme activity, efficiency, or something else. It would be helpful to clarify this.

3. Has the author performed experiments similar to what is described in Table 2 but in the presence of bicyclic ketone? I wonder if the flavin transfer occurs as fd-TCMO catalyzes. And that is why in the absence of the substrate, no difference was observed in the KmFMN even when fd-TCMO was present.

4. Has the author performed titration of FR to assess the FR-dependence of fd-TCMO activity? I think this will be helpful to understand the kinetics of FMNH2 transfer.

Author Response

I would like to thank Reviewer 1 for their positive and helpful comments.

I accept that some matters in the manuscript as submitted were inadequately explained and/or poorly presented -  getting too close to the issues to appreciate that others need to understand them in a broader perspective is a clearly mistake that needs to be corrected. Consequently, I have both made a series of significant changes to the text as originally presented, and also introduced some additional supportive material. Together, I trust these two major features of revision will have now brought greater clarity to the reworked manuscript.   

Addressing the specific issues raised:

  1. The Introduction has now been revised to highlight the two specific goals of the study as clearly indicated stand-alone sentences.

2.  and  3.  I think the Reviewer is asking two related questions here. 

In the single-enzyme kinetic studies (Line 193, not line 183) what is being measured is the the NADH + FR-dependent reduction of FMN in the absence of an aliquot of an fd-TCMO.

In the coupled-enzyme kinetic studies (Line 199), again what is being measured is the the NADH + FR-dependent reduction of FMN, but in the presence of both an aliquot of an fd-TCMO and a competent ketone substrate (rac)-bicyclo[3.2.0]hept-e-en-6-one.

In the light of the Reviewers comment, the text of entry 2.6 Couple-enzyme Kinetic Studies has been expanded to make this distinction clear.

4.  No specific investigation of this point has been made intentionally to date. However, by mistake one series of relevant kinetic studies was set up with equivalent aliquots (200 mU) of both the tested fd-TCMO preparations and the highly purified FR preparations. The resultant outcomes showed no significant differences to those shown in the recorded data included in the manuscript. Because of the nature of the discrepancy, these outcomes were not included in assessing the reproducibility the outcomes of the single-enzyme vs coupled-enzyme kinetic studies.

Aspects of the response made to Comment 4 by Reviewer 3 re conversion-dependent outcomes of the tested fd-TCMOs may also be relevant in this respect.

Reviewer 2 Report

This study confirmed the comprehensive inter-species compatibility of both native and non-native flavin reductases with each of the tested monooxygenases. For all three monooxygenases, non-native flavin reductases from Escherichia coli ATCC 11105 and Aminobacter aminovorans ATCC 29600 were confirmed to be more efficient donators of FMNH2 than the corresponding tested native flavin reductases. Some potential practical implications of these outcomes are considered for optimising FMNH2-dependent biooxy-genations of recognised practical and commercial value.  

These findings could be interested to the researchers in this area. However, some issues should be addressed by the authors. The comments were described below.

1. Purification of 2,5-DKCMO, 3,6-DKCMO, and LuxAB luciferase as well as Purification of Frp1, Frp2, FreEc, FRGVf, FreVf, and FRGAa. SDS-PAGE of these proteins should be showed in a Figure.

2. The identification of product assayed by chiral capillary GC using a Lipadex D 172 fused silica column should be provided in the Figure.

Author Response

I would like to thank Reviewer 2 for their positive and helpful comments.

I accept that some matters in the manuscript as submitted were inadequately explained and/or poorly presented -  getting too close to the issues to appreciate that others need to understand them in a broader perspective is a clearly mistake that needs to be corrected. Consequently, I have both made a series of significant changes to the text as originally presented, and also introduced some additional supportive material. Together, I trust these two major features of revision will have now brought greater clarity to the reworked manuscript.   

Addressing the specific issues raised:

  1. Confirmation relevant to the final step purification of both the fd-TCMOs and FRs has now been presented with the additional inclusion of Supplementary Figure S1A and B.

  1. Both the nature of the enantiomers of (rac)-bicyclo[3.2.0]hept-2-en-6-one and the potential lactone products resulting from their biooxidation by the fd-TCMOs has now been included in a revised version of Figure 2. These assignments can now be correlated directly with the entries (+)k, (-)k, and  (+)2l,  (-)2l, (+)3l and (-)3l in columns 3 and 4 respectively of Table 3.

Reviewer 3 Report

Dear Author,

The theme is opportune and usefull. Results are important.

However, there are some serious errors in the article, and the display of figures and tables is not always appropriate.

First, figure 2. It would be important to display the resulting products with a structural formula. The names entered do not comply with the valid entry rules. (If I assume the resulting structure correctly.) The correct names are: 3,3a,6,6a-tetrahydro-1H-cyclopenta[c]furan-1-one (instead of 3-oxabicyclo[3.2.0]oct-6-en-2-one) and 3,3a,6,6a-tetrahydro-2H-cyclopenta[b]furan-2-one ((instead of 2-oxabicyclo[3.2.0]oct-6-en-3-one). The numbers to which we refer in Table 3 should be indicated after the compounds ((+)2, (-)2; (+)3, (-)3).

Table 3 is also difficult to interpret. It is not clear which compound the given percentages ((+) and (-)) refer to? For 2 or 3? The following columns contain data either only in (-) or only in (+). If the given ratios refer to 2-oxa and 3-oxa-lactone, then this should be indicated in the header of the table.

It is also important that we do not usually indicate the "ee" value as 100%. I recommend marking >99.99.

In connection with the interpretation of Table 3, I would like to hear from you why there is such a big difference between the optical purity of the two regioisomers (2-oxa and 3-oxa-lactone)?

Author Response

I would like to thank Reviewer 3 for their positive and helpful comments.

I accept that some matters in the manuscript as submitted were inadequately explained and/or poorly presented -  getting too close to the issues to appreciate that others need to understand them in a broader perspective is a clearly mistake that needs to be corrected. Consequently, I have both made a series of significant changes to the text as originally presented, and also introduced some additional supportive material. Together, I trust these two major features of revision will have now brought greater clarity to the reworked manuscript.   

Addressing the specific issues raised:

  1. The description of the analysed outcomes of the fd-TCMO-dependent biotransformations in the submitted manuscript was very badly explained and did not correspond with the actual data presented in Tables 3. Both an appropriately modified  text and modified entries in Table 3 giving greater clarity has now been drafted into the manuscript.

  1. The nature of the lactone products resulting from the biooxidation of (rac)-   bicyclo[3.2.0]hept-2-en-6-one by the three fd-TCMOs has now been included in a revised version of Figure 2. These assignments can now be correlated directly with the entries (+)k and (-)k in column 3 of Table 3 and (+)2l, (-)2l, (+)3l and (-)3l in column 4 of Table 3. 

  1. The use of ee 100% in the manuscript for a few of the recorded lactones was included specifically in an attempt to indicate that there was no detectable level of the relevant corresponding enantiomeric lactone above ‘background noise’ in the GC output data. I have amended the relevant entries as advised.

  1. Originating from its potential to generate synthons of value for the chemoenymatic synthesis of prostaglandin analogues, (rac)-bicyclo[3.2.0] hept-2-en-6-one has been a focus of research attention over the last 30+ years.  Consistent differences in optical purity between the two regioisomeric lactones resulting from the biooxidation of the bicyclic ketone have been observed several times, by a number of different research groups, testing samples of both native and overexpressed 2,5-DKCMO, 3,6-DKCMO and LuxAB luciferase. Equivalent outcomes have also been recorded with other true flavoprotein Baeyer-Villiger monooxygenases challenged with this bicyclic ketone - these include cyclohexanone monooxygenase, cyclopentanone monooxygenase and phenylacetone monooxygenase. To date, no fully supported explanation has been advanced for the often idiosyncratic outcomes recorded, although the data from some relevant studies indicate that there are significant differences in both the speed and aspects of the selectivity with which the two ketone enantiomers are biooxidised, contributing to patterns of 2-oxa- and 3-oxa-lactone generation that are consequently conversion-dependent. If correct, the consistency of the data obtained within each individual fd-TCMO series in the current research suggests that FNMH2 supply by the range of FRs tested was not a rate-limiting factor under the conditions studied. In this particular respect, this latter part of the response may also be relevant to Comment 4 made by Reviewer 1

Round 2

Reviewer 1 Report

Thank you for addressing my comments. The manuscript has greatly improved after revision and is deemed ready for publication on my end.

Reviewer 2 Report

The authers have revised the MS based on the suggestions.